# Can the Dose Constraints Be Trusted? Actual Dose Exposure of Bladder and Rectum During Prostate Cancer Radiotherapy

**DOI:** 10.3390/cancers17071194

**Published:** 2025-03-31

**Authors:** Marc Petrikowski, Kai Kröger, Julian Roers, Dominik Hering, Sebastian Lohmann, Sergiu Scobioala, Uwe Haverkamp, Hans Theodor Eich

**Affiliations:** Department of Radiation Oncology, University Hospital Muenster, Albert-Schweitzer-Campus 1, 48149 Muenster, Germanyhans.eich@ukmuenster.de (H.T.E.)

**Keywords:** prostate cancer, radiotherapy of the prostate, image-guided radiotherapy (IGRT), adaptive radiotherapy, cone-beam computed tomography (CBCT), dose constraints

## Abstract

In prostate cancer radiotherapy, the actual and planned dose distribution differ due to daily positional and volumetric changes in the patient’s body and organs at risk. This study analyzes the deviation of the actual dose exposure from the planned dose distribution by performing dose calculations for 821 cone-beam computed tomo-graphy scans during radiotherapy. The dosimetric evaluation revealed that conventional dose evaluation via a summation plan underestimates daily dosimetric parameters. Our data suggests that daily adaptive radiotherapy should be constrained by volumetric parameters, while new dose constraints for adaptive radiotherapy must be defined based on “plan of the day” summation plans.

## 1. Introduction

Prostate cancer (PCa) is the most common malignancy in men, with 1.467 million new cases worldwide in 2022 [1]. Depending on stage, PCa can be treated with surgery, radiotherapy (RT), or active surveillance. Regarding RT, external beam radiotherapy (EBRT) and brachytherapy are available as treatment options. In EBRT, intensity-modulated radiotherapy (IMRT) offers a highly conformal dose distribution in the target volume while simultaneously sparing surrounding organs at risk (OAR) [2]. In IMRT, treatment planning is based on a planning computed tomography (pCT), which is also used for dosimetric evaluation.

Due to its anatomical location in the pelvis, the prostate experiences positional variability, leading to inaccuracies in each RT fraction during treatment [3]. Inaccuracies arise from factors such as patient setup errors, changes in bladder and rectum filling, and the resulting prostate displacement and deformation [4,5]. The establishment of image-guided radiation therapy (IGRT) using cone-beam computed tomography (CBCT) helps to identify these factors and reduce daily inaccuracies by matching pCT and CBCT. However, volume fluctuations in bladder and rectum during treatment can alter daily OAR doses despite the use of IGRT [6,7].

Previous studies demonstrated that CBCT scans can be used for dose calculations [6,8]. Since a CBCT is obtained daily according to our standard operating procedure (SOP), it is possible to calculate the actual OAR dose distribution for each fraction during RT using CBCT-based dose calculation.

This study aimed to investigate the deviation of the actual OAR dose exposure from the pCT-based dose distribution during IGRT of PCa. The impact of OAR volumes on the daily dose distribution as well as a correlation with clinical toxicities were also evaluated. By critically analyzing the current standard of care, we identified possible implications for useful dose constraints in daily adaptive RT.

## 2. Materials and Methods

### 2.1. Patients and Treatment

In this retrospective study, a total of 821 CBCT scans from 20 patients with localized PCa (T1–3, N0, M0) treated with daily IGRT at our department were included. Patients were selected chronologically since 2021 and patients with insufficient daily imaging during treatment were excluded. Additionally, patients with bilateral total hip replacement (THP) were excluded due to high artifact formation during imaging. This analysis was approved by the institutional review board (Ethikkommission der Ärztekammer West-falen-Lippe; protocol code 2024-394-f-S; 19 July 2024).

Twelve patients underwent RT with a total prescribed dose of 80 Gy in 2 Gy fractions, while eight patients received 79.2 Gy in 1.8 Gy fractions. All patients received 50/50.4 Gy to the primary planning target volume (PTV1) followed by a boost (PTV2), resulting in a cumulative dose of 80/79.2 Gy. The first clinical target volume (CTV1) included the prostate and the seminal vesicles bases (Gleason-Score ≤ 7a) or the entire seminal vesicles (Gleason-Score ≥ 7b). The boost volume (CTV2) was based on magnetic resonance imaging (MRI) and included only the prostate with a maximum margin of 0.5 mm, excluding the rectal wall. Isometric margins of 3–5 mm were formed around CTV1 and CTV2 to create PTV1 and PTV2.

Each patient underwent a pCT scan (Toshiba Acquilion LB) in treatment position (supine, arms above head, full bladder, emptied rectum) using 3 mm slice thickness. Individual treatment plans were created using the treatment planning system (TPS) Varian Eclipse version 11.0 (Varian Medical Systems, Palo Alto, CA, USA). IMRT was performed using the Varian Halcyon linear accelerator. Delineation of target volumes and OAR were performed by a radiation oncologist and treatment planning was completed according to ICRU-report 83 [9]. Dose-volume constraints used in treatment planning for the rectum based on the QUANTEC data for conventional fractioned RT were *V*50 < 50%, *V*60 < 35%, *V*65 < 25%, *V*70 < 20%, and *V*75 < 15% [10]. Constrains for the bladder were set based on the Radiation Therapy Oncology Group’s 0415 study at *V*65 < 50% and *V*70 < 35% [11].

CBCT imaging was performed before each fraction in treatment position with 3 mm slice thickness, 49.1 cm scan diameter, 430 × 430 mm resolution and half-fan filter. CBCT and pCT were matched automatically using soft-tissue registration, visually inspected by trained radiotherapists and adjusted when necessary.

CBCT scans were reviewed offline by a trained physician assessing OAR filling and position. Ahead of treatment, all patients were given clear recommendations for optimal organ filling for the pCT and for each RT fraction. Regarding the bladder filling, patients were instructed not to use the lavatory for at least 30 min before treatment and to drink at least 300 mL of water during this time. For optimal rectum filling patients were instructed not to eat any flatulent foods and to empty the rectum. In case of severely differing OAR volumes, laxatives and an optimized registration or a new pCT scan was ordered. All patients underwent regular visits from the treating physician to monitor treatment-related toxicities, intervening when necessary.

### 2.2. Dosimetric Evaluation Using CBCT

To calculate the actual dose distribution based on each CBCT as precisely as possible, the gray scale values of the CBCT scans had to be converted to physical density. An established region of interest (ROI) CT number mapping method was used to create a CBCT-number to physical-density calibration curve, enabling the calculation of dose distributions based on daily CBCT data sets [6,8]. This method was applied to 182 pelvic CBCT scans, acquired under the conditions mentioned above, from 26 patients with localized PCa. The method comprises the following steps:CT and CBCT images were registered in the TPS and matched automatically.ROIs were mapped from CT to CBCT and mean Hounsfield units were measured in both images. ROIs included both femoral heads, bladder, muscle, fatty tissue and air.The physical density calibration curve was generated based on CT density values.

Prostate and OAR were contoured for each CBCT scan using Limbus Contour v.1.8.0 (Limbus AI Inc., Regina, SK, Canada). Correctness of organ-contouring was verified manually and adjusted if necessary. Individual treatment plans were copied to each CBCT data set, and a daily dose recalculation was performed for all 821 CBCT scans.

Daily dosimetric parameters such as maximum dose (*D*_max_), mean dose (*D*_mean_) and volume were extracted from each fraction’s dose-volume histogram (DVH). To assess whether treatment planning constraints were met daily, dosimetric parameters were adjusted relative to the daily fraction dose using the following formula:(1)$$\frac{Dose\;Constraint}{Total\;Dose}\;\cdot Daily\;Dose=Daily\;Dose\;Constraint$$

Formula (1) was used to calculate bladder *V*65_Daily_ and *V*70_Daily_ and rectum *V*50_Daily_, *V*60_Daily_, *V*65_Daily_ and *V*70_Daily_. $\bar{V}50$_Daily_, $\bar{V}60$_Daily_, $\bar{V}65$_Daily_, $\bar{V}70$_Daily_, $\bar{D}$_max_ and $\bar{D}$_mean_ correspond to the mean values of the daily parameters over all CBCT scans.

Two different methods were used to assess the deviation between actual and planned OAR dose exposure. Firstly, a summation plan was calculated, by incorporating all CBCT volume structures during a patient’s treatment into the pCT. That CBCT-Volume summation plan was then compared to the initial summation plan. Secondly, each daily CBCT fraction was analyzed individually and its daily dosimetric parameters were compared with those of the initial summation plan, which was adjusted relative to the daily fraction dose.

### 2.3. Correlation with Follow-Up Data

Patients were offered regular follow-up examinations to assess possible acute and late toxicities. Patient characteristics and clinical data were collected from the hospital information system (Orbis, Dedalus Health Care, Bonn, Germany). Data collection ended in June 2024. Recorded toxicities during and after treatment were classified according to National Cancer Institute’s Common Terminology Criteria for Adverse Events v.5.0 [12].

Patients were divided dichotomously based on grades of the registered toxicities, investigating whether increased dosimetric parameters were associated with increased toxicities. Parameters for bladder and rectum were examined separately with toxicities in the corresponding organ system.

### 2.4. Statistical Analysis

Each fraction’s DVH was analyzed using RStudio v.2024.04 (PBC, Boston, MA, USA). All further statistical analyses were performed using SPSS^®^ v.29.0.2.0 (IBM^®^, Armonk, NY, USA). *p* values less than 0.05 were considered statistically significant.

Prior to statistical testing, each variable was analyzed for normal distribution using the Kolmogorov-Smirnov test. For correlation analysis, Pearson and Spearmen correlation tests were performed for normally and non-normally distributed variables, respectively. The paired *t*-test and the Wilcoxon signed-rank test were used for continuous, paired, normally and non-normally distributed variables. Finally, the Mann-Whitney U test was performed to investigate whether increased daily dosimetric parameters were measured in patients with higher-grade toxicities.

## 3. Results

### 3.1. Dosimetric Evaluation

A total of 20 patients with localized PCa were analyzed. Patient baseline characteristics are shown in Table A1. Out of 821 CBCT scans, 778 (94.8%) were correctly contoured for plan evaluation, while 43 scans (5.2%) from an obese patient with unilateral hip replacement required manual contouring due to imaging artifacts.

CBCT-based plan calculation enabled a dosimetric evaluation of each individual treatment fraction, exemplified by two fractions from the same patient (Figure 1a–d). Daily positional and volumetric changes influenced the associated DVH and dosimetric parameters (Figure 1e,f).

DVH examination for each fraction revealed a fluctuating divergence from planned values (Figure 2a,b). In the patient shown, daily bladder *D*_mean_ fluctuated between 19% to 83% of the daily dose; rectum *D*_mean_ fluctuated between 30% to 79% of the daily dose.

Figure 2c,d demonstrate the average deviation of daily parameters from planned values (CBCT/plan) across all 821 fractions. The actual *D*_mean_ for both bladder and rectum exceeded planned values. The median deviation (IQR) was 1.11 (0.89–1.33) for bladder *D*_mean_ and 1.16 (1.02–1.31) for rectal *D*_mean_. Daily volumetric parameters for both bladder and rectum were all higher than planned values. The deviation increased with the size of the respective parameter.

Table 1 compares the average dosimetric parameters of the initially planned summation plan with both the CBCT summation plan based on daily matching of pCT and CBCT, and the parameters calculated based on daily CBCT evaluation. Comparing the initial and CBCT summation plans, all bladder and rectum parameters are, on average, lower in the CBCT summation plan than assumed during treatment planning. However, only rectum $\;\bar{V}50$_Daily_ and $\bar{D}$_max_ were significantly reduced.

Daily CBCT evaluation in comparison to planned values showed a significant increase in bladder $\bar{V}65$_Daily_ and $\bar{V}70$_Daily_ at significantly smaller daily bladder volumes. Average CBCT values for all rectal volumetric parameters as well as $\bar{D}$_mean_ were significantly increased, while average daily rectal volumes were significantly smaller.

### 3.2. Dose-Volume Relationship

Interfractional changes in OAR volumes are shown in Figure 3a,b. During treatment, bladder and rectum volumes differed from planned organ volumes in every fraction, while variation in absolute bladder volumes was significantly higher than in rectal volumes (Table A2, *p* < 0.001). However, there was no significant difference in the relative deviation from planned volumes.

Increased bladder volumes correlated significantly with decreased bladder *D*_mean_ (*r* = −0.553, *p* < 0.001) and decreased V65–70_Daily_ (*r* = −0.489, *r* = −0.473; *p* < 0.001 each). However, no fitting regression model was found for absolute volumes (Figure 3c). Relatively to planned values, increased bladder volumes correlated with decreased relative *D*_mean_ (*r* = −0.782, *p* < 0.001) and decreased relative V65–70_Daily_ (*r* = −0.512, *r* = −0.461, *p* < 0.001 each). The relationship between relative bladder volumes and *D*_mean_ fitted an exponential model in non-linear regression (Figure 3d). Increased bladder *D*_mean_ correlated with increased *V*65_Daily_ and *V*70_Daily_ values (*r* = 0.924, *r* = 0.909, *p* ≤ 0.001 each). These relationships also matched exponential models (Figure 3e,f). 

Increased rectal volumes correlated with increased *D*_mean_ and *D*_max_ (*r* = 0.074, *p* = 0.033; *r* = 0.169, *p* < 0.001). However, no suitable regression model was found for absolute or relative values (Figure 4a,b). Increased rectum *D*_mean_ correlated with increased *V*50–70_Daily_ (*r* = 0.945, *r* = 0.909, *r* = 0.88, *r* = 0.843; *p* < 0.001 each). These relationships matched linear regression models (Figure 4c–f).

### 3.3. Correlation Between Toxicity and Dose

Reported genitourinary (GU) and gastrointestinal (GI) toxicities are shown in Table A3. Median follow-up time from start of RT was 18 months (range: 2–27 months). During the entire observation period, GI or GU toxicity ≥ grade 2 was recorded in 4 patients (20%) each. Table 2 demonstrates a significantly increased bladder *D*_mean_ in patients with GU toxicity ≥ grade 2. Furthermore, patients with increased GU toxicities showed on average non-significantly increased *V*65–70_Daily_ at smaller bladder volumes. In patients with increased GI toxicities, rectal *V*50–70_Daily_ and volume were non-significantly increased.

## 4. Discussion

This study compared planned OAR dose distribution for IGRT in PCa with the actual OAR dose exposure through CBCT-based daily dose calculation. It also explored the impact of OAR volumes on the daily dose distribution and their correlation with clinical toxicities.

Daily dosimetric evaluation revealed a significant increase in all volumetric parameters for both the bladder and the rectum compared to the planned values. In contrast to the daily bladder *D*_mean_, the daily rectum *D*_mean_ was significantly increased. An exponential dose-volume relationship was observed for the bladder, but no definitive dose-volume relationship could be shown for the rectum. Overall, RT was tolerated well. Patients with increased GU toxicities generally had higher volumetric parameters at smaller bladder volumes, while those with increased GI toxicities had higher volumetric parameters at larger rectum volumes.

### 4.1. How Reliable Is a Summation Plan?

OAR total dose evaluation is of particular interest as a possible quality control instrument. In IGRT, the pCT is assumed to represent the entire treatment, assuming that OAR volumes during treatment follow a roughly normal distribution and that GTV shifts fall within the PTV margin [13]. The pCT’s DVH serves as the baseline for dosimetric evaluation, since adherence to dose constraints directly impacts organ toxicities [10,11]. However, daily dosimetric evaluation showed a notable discrepancy between actual and planned parameters, despite the use of IGRT. While IGRT can correct possible positioning errors, it cannot account for positional or volumetric changes [14], resulting in fractions where measured parameters exceed defined constraints.

Two methods were presented for the comparison of actual and planned OAR dose exposure. When creating a CBCT-based summation plan, all volume structures of individual fractions are simply incorporated into the pCT. However, as this method only projects anatomical changes onto the original plan, no clear conclusion can be drawn for the daily dose distribution, apart from the statement that OAR matching worked well. Due to the averaging of OAR volumes, the method is less accurate than an analysis of each fraction.

Daily CBCT evaluation showed that pCT-based summation plan analyses underestimate daily volumetric parameters. Large volumetric parameters like *V*65_Daily_ or *V*70_Daily_ are highly dependent on volume changes and thus deviate the most from planned values (Figure 2) [15]. Particularly the rectum can quickly shift into the PTV, due to the proximity of the anterior rectal wall. These deviations are only represented to a limited extent by the respective *D*_mean_, as shown for the bladder where increased *V*65–70_Daily_ were not associated with increased *D*_mean_. This is important for the development of meaningful constraints for “plan of the day” (POTD) adaptive RT. When the current generation of devices was introduced in our clinic, we considered which daily constraints are most important for a POTD SOP. Our analyses suggest that volumetric parameters like *V*65_Daily_ and *V*70_Daily_ should be used rather than *D*_mean_ or *D*_max_ values.

Despite the observed deviations from the original treatment plan, all established dose constraints set for bladder and rectum during RT planning were still met on average, with only rectum $\bar{V}65$_Daily_ and $\bar{V}70$_Daily_ being minimally exceeded. However, known dose constraints are not based on summed up daily CBCT analyses, but on summation plan analyses based on single pCT scans. Concluding that established EBRT constraints cannot be reduced to a single fraction, new constraints must be defined for adaptive RT that are not based on conventional, but on POTD summation plans.

### 4.2. Impact of OAR Volumes on Dose Distribution

Despite clear patient recommendations regarding OAR filling, variances in OAR volumes are unavoidable (Figure 3a,b), whereby the variation in bladder volumes is higher than in rectum volumes as previous studies have shown [16].

Analyzing the bladder’s dose-volume relationship, Chen et al. and Fuchs et al. found a linear relationship [16,17], whereas our data showed an exponential relationship, in agreement with Roch et al. [15]. Our data support the assumption that increased bladder volumes improve organ sparing. Increased bladder volumes cause parts of the bladder to move out of the horizontal irradiation plane beyond the upper field limit. As a result, the proportion of the organ that is exposed to the high-dose area is much smaller, which could explain the exponential dose volume relationship of the bladder. Maintaining a high bladder volume provides a significant dosimetric advantage for bladder protection during RT, which should be maximized during treatment if the patient’s condition permits.

For the rectum, the dosimetric analysis is not as clear. Although increased rectum volumes influence prostate displacement and deformities, the ESTRO guideline on the use of IGRT in PCa does not provide general recommendations for rectum filling during RT [18]. Our data suggests that a minimal rectum volume improves rectum sparing. While Roch et al. reported that increased rectum volumes correlate with increased *D*_max_ and decreased *D*_mean_ [15], our data showed that increased rectum volumes correlated with both increased *D*_max_ and *D*_mean_. This seems reasonable since, with an enlarged rectum, larger parts of the organ protrude towards the prostate and thus into the high-dose area around the PTV. This mainly horizontal movement within the irradiation plane might explain the observed linear dose volume relationship of the rectum.

However, our daily CBCT evaluation showed that on average both $\bar{D}$_mean_ and $\bar{D}$_max_ were significantly increased, with significantly reduced rectum volumes. This significant reduction in rectum volumes seems to initially contradict our findings, but can be explained by the fact that the effect of rectal filling on contour and position relative to the prostate is subject to both individual and situational influences [19]. Therefore, rectal volume alone has only a limited influence on the daily dose exposure. Our analysis shows that, in addition to absolute rectum volume, the daily positioning of the rectum in relation to the PTV is decisive for the daily dose exposure.

### 4.3. Toxicities

RT was tolerated well in our cohort. Observed toxicity rates are comparable to those previously reported [20,21,22,23,24], exemplarily shown for acute toxicities in Table A4. Although comparisons are difficult due to our small cohort and differences between the presented studies regarding treatment protocols and dose constraints, observed toxicities rates are within the normal range.

Based on our dosimetric evaluation, we anticipated that patients with increased GU toxicities would show increased bladder $\bar{D}$_mean_ and volumetric parameters at low bladder volumes, while in patients with increased GI toxicities, increased rectum volumes associated with increased $\bar{D}$_mean_ and volume parameters were to be expected. Expectations for the correlation between daily dose and toxicities were met on average, but significance could only be shown for increased bladder $\bar{D}$_mean_ (Table 2).

Schaake et al. reported that GU toxicities were associated with high bladder-wall doses and GI toxicities were associated with increased rectum *V*70 [25]. Our analysis identified that adaptive RT should be constrained by daily volumetric parameters like *V*65_Daily_ and *V*70_Daily_. A daily monitoring and possible reduction of the daily dose exposure in the PTV surrounding areas could reduce the cumulative dose to the respective OAR and therefore reduce treatment-related toxicities. The impact of daily CBCT based adaptive RT on treatment-related organ toxicities in the treatment of PCa is part of current clinical investigations, however a potential reduction has been reported for MRI-guided online ART [26].

The implementation of daily CBCT scans during the IGRT of PCa raises concerns regarding the additional radiation exposure to patients [27]. We acknowledge that daily CBCT image guidance does add radiation exposure, but the benefits of an optimized positioning outweigh the risks. This conclusion is based on various studies showing that daily CBCT-based IGRT allows a reduction in PTV margins [28,29] and thus leads to lower toxicity rates [30].

Several previous studies used CBCT to investigate volume changes in bladder and rectum during treatment [6,7,15,16,31]. However, in these studies, patients either did not receive CBCT imaging before each RT fraction [16,31] or only some CBCTs were included [7]. Therefore, no entire dosimetric evaluation could be performed or a hypofractionated treatment concept was utilized [6,15]. To our knowledge, this is the first study to investigate whether conventional EBRT treatment planning constraints in RT of PCa were met daily by performing CBCT based dose calculations.

Despite the inclusion of more than 800 CBCT scans, there are limitations to this study, as it analyses retrospective data from a relatively small patient cohort. In addition, it only represents the experience of a single center. Prospective evaluation of larger patient collectives is needed to gather more data regarding daily dosimetric evaluation and to further evaluate which single-fraction dose constraints should be used for online adaptive RT. However, the study includes real-world data from the clinical routine, enabling a critical analysis of the current standard of care and establishes a link between daily dosimetric analysis and clinical toxicities.

## 5. Conclusions

Conventional dose evaluation via a summation plan underestimates daily dosimetric parameters, whereas the presented method enables a more precise evaluation of the daily dose distribution and is well-suited for retrospective quality management. In IGRT of PCa, the bladder is subject to a dose-volume relationship, while the daily positional relationship to the PTV is decisive for the dose exposure of the rectum. For adaptive RT of PCa, volumetric parameters rather than mean doses should be used for daily treatment planning constraints. Reasonable constraints should consider daily positional and volumetric changes in OAR and CTV, rather than relying on a single pCT.

## Figures and Tables

**Figure 1 cancers-17-01194-f001:**
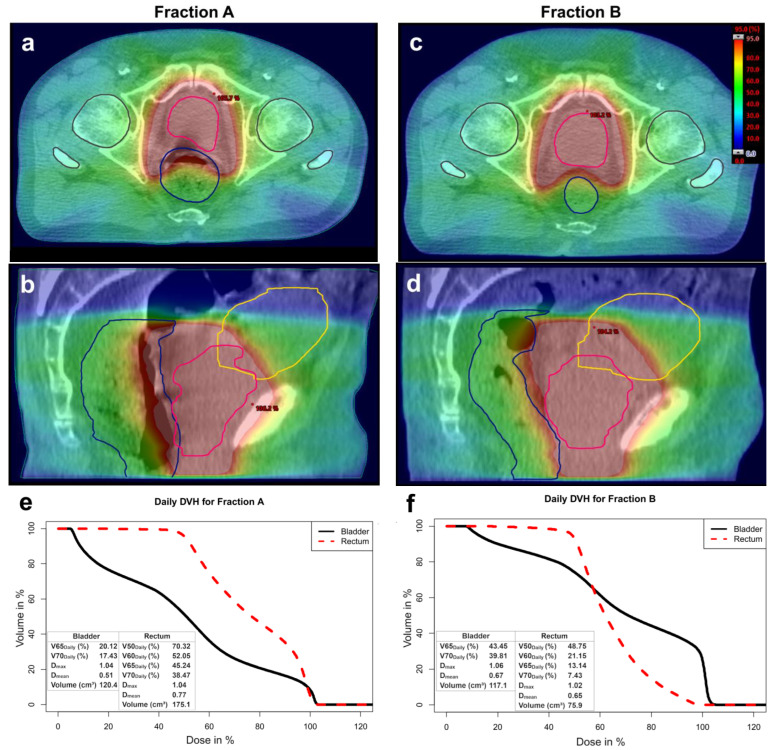
Daily dose distribution in the same transversal and sagittal plane shown exemplarily for two fractions of the same patient. Organ contours are highlighted respectively. (**a**,**b**) Daily dose distribution of fraction A; (**c**,**d**) Daily dose distribution of fraction B; (**e**) Corresponding daily DVH for fraction A; (**f**) Corresponding daily DVH for fraction B.

**Figure 2 cancers-17-01194-f002:**
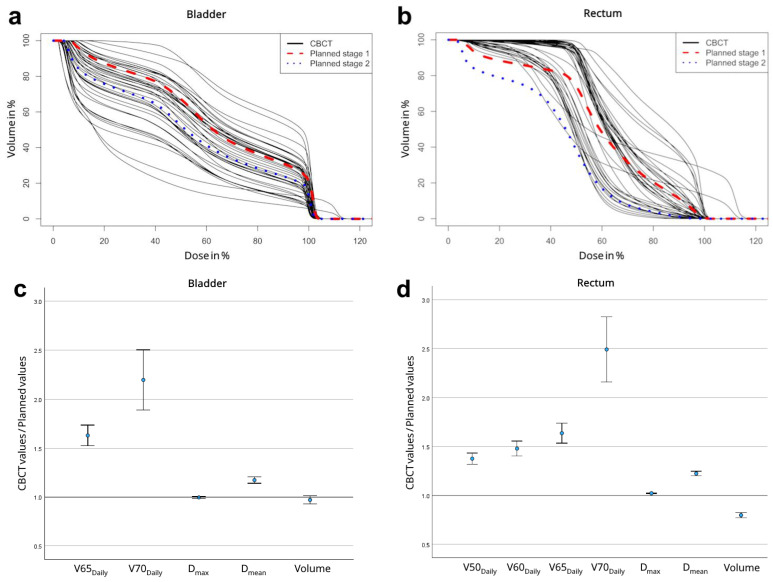
DVH of each fraction during the patient’s treatment for (**a**) bladder and (**b**) rectum. Mean deviations of CBCT values from planned values for (**c**) bladder and (**d**) rectum in all patients. Error bars correspond to 95% confidence interval.

**Figure 3 cancers-17-01194-f003:**
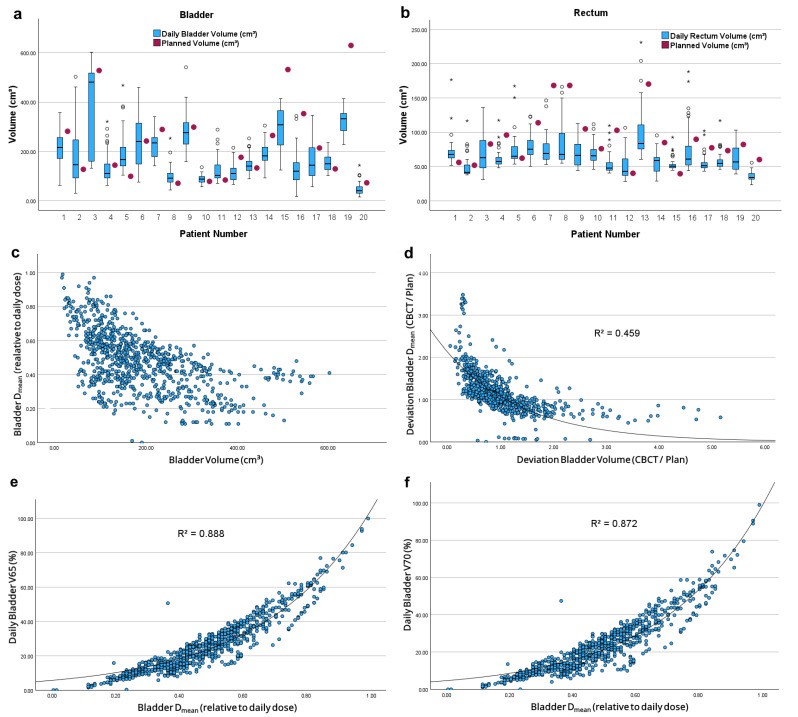
Changes in absolute (**a**) bladder and (**b**) rectum volumes compared to planned volumes; (**c**) Correlation between absolute bladder volumes and *D*_mean_ in proportion to the daily dose; (**d**) Correlation between bladder volumes and bladder *D*_mean_, each in relation to planned values; Correlation between bladder *D*_mean_ and (**e**) *V*65_Daily_ and (**f**) *V*70_Daily_. * marks extreme values. Circle marks outliers.

**Figure 4 cancers-17-01194-f004:**
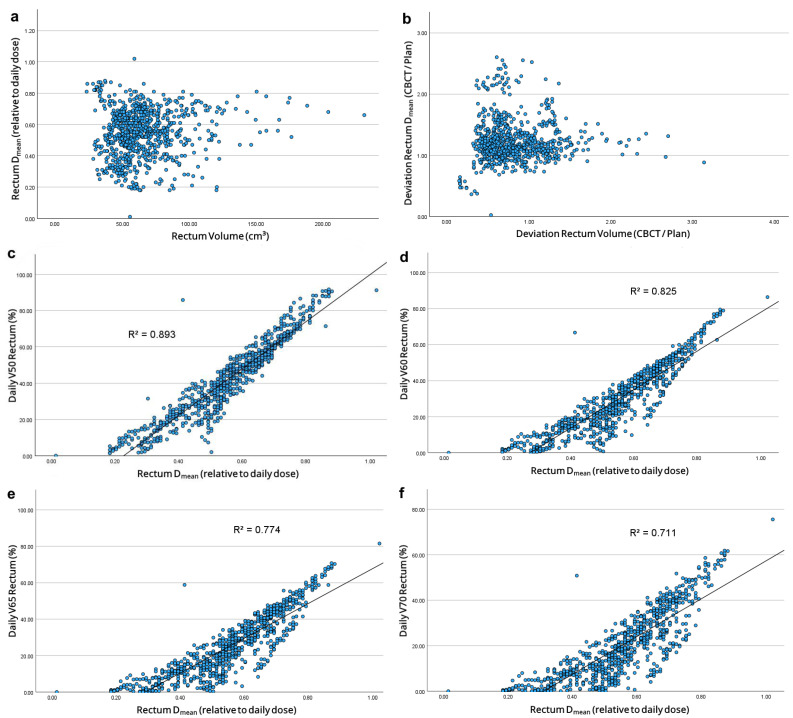
(**a**) Correlation between absolute rectum volumes and *D*_mean_ in proportion to the daily dose; (**b**) Correlation between rectum volumes and rectum *D*_mean_ each in relation to planned values; Correlation between mean rectum doses and daily volumetric parameters of the rectum: (**c**) *V*50_Daily_; (**d**) *V*60_Daily_; (**e**) *V*65_Daily_ and (**f**) *V*70_Daily_.

**Table 1 cancers-17-01194-t001:** Average values of dosimetric parameters defined during treatment planning compared with those from CBCT summation plans and those calculated based on daily CBCT scans.

	Parameter	Initial Summation Plan	CBCT Summation Plan (*n* = 20)	*p*
Bladder	$\bar{V}65$ (%)	19.01 ± 8.91	18.17 ± 8.89	0.113
	$\bar{V}70$ (%)	14.91 ± 7.81	13.74 ± 8.13	0.138
	$\bar{D}$_max_ (Gy)	81.99 ± 1.09	81.02 ± 2.95	0.184
	$\bar{D}$_mean_ (Gy)	35.36 ± 10.98	34.83 ± 10.53	0.150
Rectum	$\bar{V}50$ (%)	32.17 ± 12.34	30.71 ± 12.04	0.021
	$\bar{V}60$ (%)	19.80 ± 9.98	18.75 ± 10.15	0.114
	$\bar{V}65$ (%)	14.44 ± 8.51	13.40 ± 8.74	0.139
	$\bar{V}70$ (%)	9.17 ± 6.80	8.20 ± 7.02	0.170
	$\bar{D}$_max_ (Gy)	79.36 ± 1.09	78.19 ± 2.82	0.040
	$\bar{D}$_mean_ (Gy)	36.95 ± 6.56	36.45 ± 6.29	0.107
	**Parameter**	**Initial ** **Summation Plan**	**Daily CBCT** **(*n* = 821)**	** *p* **
Bladder	$\bar{V}65$_Daily_ (%)	19.01 ± 8.91	25.96 ± 11.43	0.003
	$\bar{V}70$_Daily_ (%)	14.91 ± 7.81	23.07 ± 10.74	<0.001
	$\bar{D}$_max_ *	1.04 ± 0.04	1.05 ± 0.02	0.041
	$\bar{D}$_mean_ *	0.45 ± 0.14	0.48 ± 0.13	0.246
	Volume (cm^3^)	237.3 ± 164.9	186.8 ± 85.5	0.042
Rectum	$\bar{V}50$_Daily_ (%)	32.17 ± 12.34	42.06 ± 14.61	<0.001
	$\bar{V}60$_Daily_ (%)	19.80 ± 9.98	30.07 ± 11.65	<0.001
	$\bar{V}65$_Daily_ (%)	14.44 ± 8.51	25.04 ± 10.36	<0.001
	$\bar{V}70$_Daily_ (%)	9.17 ± 6.80	20.12 ± 9.20	<0.001
	$\bar{D}$_max_ *	1.01 ± 0.04	1.03 ± 0.02	0.016
	$\bar{D}$_mean_ *	0.47 ± 0.09	0.55 ± 0.11	<0.001
	Volume (cm^3^)	86.6 ± 35.3	64.8 ± 14.3	0.002

* Daily dose parameters are given as a proportion of the dose per fraction.

**Table 2 cancers-17-01194-t002:** Average daily dosimetric parameters based on CBCT calculation comparing patients with toxicities ≥ grade 2 and patients with toxicities ≤ grade 1.

	Parameter	Genitourinary Toxicity Grade ≤ 1 (*n* = 16)	Genitourinary Toxicity Grade ≥ 2 (*n* = 4)	*p*
Bladder	$\bar{V}65$_Daily_ (%)	23.70 ± 9.44	35.01 ± 15.66	0.219
	$\bar{V}70$_Daily_ (%)	21.14 ± 8.93	30.78 ± 15.24	0.299
	$\bar{D}$_max_ *	1.06 ± 0.02	1.04 ± 0.01	0.059
	$\bar{D}$_mean_ *	0.44 ± 0.10	0.63 ± 0.13	0.014
	Volume (cm^3^)	209.3 ± 79.1	96.5 ± 37.9	0.080
	**Parameter**	**Gastrointestinal Toxicity** **Grade ≤ 1** **(*n* = 16)**	**Gastrointestinal Toxicity** **Grade ≥ 2** **(*n* = 4)**	* **p** *
Rectum	$\bar{V}50$_Daily_ (%)	42.05 ± 15.65	42.09 ± 11.24	0.777
	$\bar{V}60$_Daily_ (%)	29.90 ± 12.62	30.78 ± 7.82	0.508
	$\bar{V}65$_Daily_ (%)	24.76 ± 11.19	26.16 ± 7.13	0.571
	$\bar{V}70$_Daily_ (%)	19.77 ± 9.86	21.55 ± 6.88	0.571
	$\bar{D}$_max_ *	1.03 ± 0.02	1.03 ± 0.03	0.701
	$\bar{D}$_mean_ *	0.55 ± 0.11	0.55 ± 0.04	0.962
	Volume (cm^3^)	63.9 ± 15.3	68.6 ± 10.5	0.508

* Daily dose parameters are given as a proportion of the dose per fraction.

## Data Availability

Data relevant to this study are presented in the paper. Public deposition of data is not possible due to restrictions put in place by the Institutional Review Board.

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
