# Peer review of "Can the Dose Constraints Be Trusted? Actual Dose Exposure of Bladder and Rectum During Prostate Cancer Radiotherapy"

_cancers, 2025, doi:10.3390/cancers17071194_

Round 1
Reviewer 1 Report
Comments and Suggestions for Authors
Title:
Can the dose constraints be trusted? Actual dose exposure of bladder and rectum during prostate cancer radiotherapy
Interesting retrospective study conducted on 821 CBCT during radiotherapy in patients with prostate cancer undergoing curative normofractionated radiotherapy treatment.
The study well shows the movement of the rectum and bladder during radiation treatment potentially affecting the dosimetry of the plain.
Highlights of the study:
1 Current topic of interest
2 The tables have been well constructed, to facilitate the interpretation for the user.
Weaknesses of the study:
1 Retrospective experience
Request for revision:
A Can you investigate the rectal and bladder preparation used?
B Is systematic use of bladder scan in monitoring bladder preparation likely to improve bladder reproducibility? has it been used?
C Has rectal preparation been optimised with the use of rectal enemas during radiation treatment?
Author Response
Dear reviewer,
thank you very much for your feedback. We appreciate your support and expertise. Please find the detailed responses to your comments below.
Comment 1: Can you investigate the rectal and bladder preparation used?
Response 1: Ahead of treatment, all patients were given clear recommendations for optimal organ filling for the planning computed tomography and for each radiotherapy fraction. Regarding the bladder filling patients were instructed not to use the lavatory for at least 30 minutes before treatment and to drink at least 300 ml of water during this time. For optimal rectum filling patients were instructed not to eat any flatulent foods and to empty the rectum. In case of severely differing volumes, laxatives and an optimized registration or a new pCT scan was ordered.
For clarification we have added a paragraph on the rectal and bladder preparations used (please see page 3, lines 102-108). As discussed, variances in OAR volumes were unavoidable, despite clear patient recommendations regarding OAR filling (Figure 3a-3b; page 11, line 288-290). Because radiotherapy was well tolerated in our cohort and the evaluation of the interfractional changes in OAR volumes showed that variances in OAR volumes are subject to individual and situational influences (lines 310-317), we conclude that the described rectal and bladder preparations generate appropriate OAR volumes.
Comment 2: Is systematic use of bladder scans in monitoring bladder preparation likely to improve bladder reproducibility? Has it been used?
Response 2: Because IGRT was performed on daily pelvic CBCT-scans, a monitoring of the daily bladder volume was possible for all 821 radiotherapy fractions included in this study. The daily assessment of each fraction’s bladder-volume improved bladder reproducibility not only due to the offline review but also due to the direct feedback to the patient after each fraction.
Comment 3: Has rectal preparation been optimized with the use of rectal enemas during radiation treatment?
Response 3: Apart from the measures for rectal preparation described above, no additional preparations like rectal enemas were used.
We hope that the proposed revisions have effectively addressed your comments. Should there be any further inquiries or issues that require clarification, we would be more than happy to provide additional information.
With kind regards on behalf of the authors
Marc Petrikowski
Kai Kröger
Hans Theodor Eich
Reviewer 2 Report
Comments and Suggestions for Authors
The title perfectly summarizes the spirit of the work, exploring not only exclusively dosimetric evaluations (doses to the rectum and bladder) but also correlating them to the toxicity developed by the patients. The entire methodology is clearly described. The statistical analysis is accurate and well described. The results are detailed: the tables, figures and graphs are a fundamental part of the work that make it quickly understandable and usable. Results are properly supported by comparison with the literature (both when they are aligned and when they are in contrast with the available literature). The bibliography is consistent with the work. The study has some limitations: retrospective analysis, few patients and conventional fractionation (which is progressively less used) but these limitations have been declared by the authors themselves.
Possible typo on lines 110 and 111 regarding CBCT and patient number.
To make the article more easily understandable to clinical readers, I have a minor suggestion: the two cumulative dose evaluation systems (planned summation plan vs CBCT summation plan) might be better explained in their methodology and their comparison inside the “Materials and methods” at paragraph 2.2 "Dosimetric evaluation".
Author Response
Dear reviewer,
thank you very much for taking the time to review this manuscript. Your thoughts certainly helped to elaborate the script. Please find the detailed responses to your comments below.
Comment 1: Possible typo on lines 110 and 111 regarding CBCT and patient number.
Response 1: We confirm that the CBCT-number to physical-density calibration curve was created based on 182 pelvic CBCT scans from 26 patients with localized prostate cancer. So, from each patient seven CBCT scans were included. 140 CBCT scans belonged to the 20 patients for whom we performed a complete daily dose calculation, an additional 42 CBCT scans of 6 other patients were included, only for the creation of the pelvic calibration curve. All CBCT scans were acquired under the same conditions so that a calibration curve was created according to the methodology presented by Richter et al. (Reference 8).
Comment 2: To make the article more easily understandable to clinical readers, I have a minor suggestion: the two cumulative dose evaluation systems (planned summation plan vs CBCT summation plan) might be better explained in their methodology and their comparison inside the “Materials and methods” at paragraph 2.2 "Dosimetric evaluation".
Response 2: We thank you for your suggestion and have added a paragraph explaining the two methods under paragraph 2.2. (please see page 3f., lines 133-139). In addition, we have renamed the different summation plans for clarification (please see page 6, lines 187-191 and page 7, Table 1).
We hope that the proposed revisions have effectively addressed your comments. Should there be any further inquiries or issues that require clarification, we would be more than happy to provide additional information.
With kind regards on behalf of the authors
Marc Petrikowski
Kai Kröger
Hans Theodor Eich
Reviewer 3 Report
Comments and Suggestions for Authors
This study investigates the deviation between planned and actual dose distributions to the bladder and rectum during image-guided radiotherapy (IGRT) for prostate cancer, using daily cone-beam computed tomography (CBCT) scans from 20 patients. The findings reveal that conventional dose evaluation underestimates daily dosimetric parameters, with significant increases in bladder and rectum doses compared to planned values. An exponential dose-volume relationship was observed for the bladder, while rectal dose exposure was influenced by daily positional changes. Patients with higher genitourinary toxicities received increased bladder doses, suggesting that adaptive radiotherapy should prioritize volumetric parameters over mean doses. The study concludes that new dose constraints should account for daily anatomical variations rather than relying on a single planning CT scan.
Page 1, "This study analyzes the deviation of the actual dose exposure from the planned dose distribution, by performing a dose calculation for 821 cone-beam computed tomography scans during radiotherapy."You can change : "This study analyzes the deviation of the actual dose exposure from the planned dose distribution by performing dose calculations for 821 cone-beam computed tomography scans during radiotherapy."
Page 2, Introduction: "Due to its anatomical location in the pelvis, the prostate experiences positional variability, causing inaccuracies for each RT fraction during treatment [3]." You can change : "Due to its anatomical location in the pelvis, the prostate experiences positional variability, leading to inaccuracies in each RT fraction during treatment [3]."
Page 3, Materials and Methods: "Twelve patients underwent RT with a total prescribed dose of 80 Gy in 2 Gy fractions, eight patients had RT with 79.2 Gy in 1.8 Gy fractions." You can change: "Twelve patients underwent RT with a total prescribed dose of 80 Gy in 2 Gy fractions, while eight patients received 79.2 Gy in 1.8 Gy fractions."
Page 4, Dosimetric evaluation using CBCT: "To calculate the actual dose distribution based on every single CBCT as precisely as possible, gray scale values of CBCT scans had to be converted to a physical density." You can change : "To calculate the actual dose distribution based on each CBCT as precisely as possible, the gray scale values of the CBCT scans had to be converted to physical density."
Page 5, Results: "Daily positional and volumetric changes influenced the associated DVH and dosimetric parameters (Figure 1e-1f)." Youcan change : "Daily positional and volumetric changes influenced the associated dose-volume histograms (DVH) and dosimetric parameters (Figure 1e-1f)."
Page 6, Results: "The median deviation (IQR) was 1.11 (0.89-1.33) for bladder Dmean and 1.16 (1.02-1.31) for rectum Dmean." Youcan change : "The median deviation (IQR) was 1.11 (0.89-1.33) for bladder Dmean and 1.16 (1.02-1.31) for rectal Dmean."
Page 7, Results: "Comparing planned and CBCT summation plan, all bladder and rectum parameters are lower on average in the CBCT summation plan than assumed in treatment planning." You can change : "Comparing the planned and CBCT summation plans, all bladder and rectal parameters are, on average, lower in the CBCT summation plan than assumed during treatment planning."
Page 8, Results: "Increased bladder volumes correlated significantly with decreased bladder Dmean (r=−0.553,p<0.001) and decreased V65−70Daily (r=−0.489,r=−0.473;p<0.001 each)." You can change : "Increased bladder volumes correlated significantly with decreased bladder Dmean (r=−0.553,p<0.001) and decreased V65−70Daily (r=−0.489,r=−0.473;p<0.001 each)."
Page 9, Results: "Increased rectum volumes correlated with increased Dmean and Dmax (r=0.074,p=0.033;r=0.169,p<0.001)." You can change :"Increased rectal volumes correlated with increased Dmean and Dmax (r=0.074,p=0.033;r=0.169,p<0.001)."
Page 10, Discussion:"Daily dosimetric evaluation revealed a significant increase in all volumetric parameters for both bladder and rectum compared to planned values." You can change :"Daily dosimetric evaluation revealed a significant increase in all volumetric parameters for both the bladder and rectum compared to the planned values."
-The study includes only 20 patients, which is a relatively small sample size. This limits the statistical power and generalizability of the findings, as the results may not be representative of a broader population.
-The study focuses primarily on dosimetric parameters (e.g., V65, V70) and their correlation with toxicities but does not provide data on clinical outcomes such as cancer control rates, progression-free survival, or overall survival.
Author Response
Dear reviewer,
thank you very much for your constructive criticism and your ideas on how to improve the quality of the text. We have adopted the language changes you suggested.
Regarding your comments, please find the detailed responses below.
Comment 1: The study includes only 20 patients, which is a relatively small sample size. This limits the statistical power and generalizability of the findings, as the results may not be representative of a broader population.
Response 1: We agree with your comment and acknowledge that despite the inclusion of more than 800 CBCT scans the generalizability of this study is limited by factors such as its retrospective design and small sample size. To address this comment, we have modified our remarks regarding the limitations of this study (please see page 12, lines 352-359).
Comment 2: The study focuses primarily on dosimetric parameters (e.g., V65, V70) and their correlation with toxicities but does not provide data on clinical outcomes such as cancer control rates, progression-free survival, or overall survival.
Response 2: We appreciate your suggestion and agree with the concept that dosimetric evaluation must be correlated with clinical outcome. But as the presented study analysis the actual dose exposure of the organs at risk bladder and rectum it investigated the toxicity rates in the respective organ system as a clinical outcome. In future studies a similar daily CBCT-based method could be used to investigate the actual dose coverage of the CTV by the PTV. The impact of the actual dose coverage on the oncological results could then also be investigated.
We hope that the proposed revisions have effectively addressed your comments. Should there be any further inquiries or issues that require clarification, we would be more than happy to provide additional information.
With kind regards on behalf of the authors
Marc Petrikowski
Kai Kröger
Hans Theodor Eich
Reviewer 4 Report
Comments and Suggestions for Authors
In manuscript ‘Can the dose constraints be trusted? Actual dose exposure of bladder and rectum during prostate cancer radiotherapy’, The author utilizes daily CBCT images acquired through IGRT in prostate cancer patients to overlay the treatment plan onto these images and recalculate the dose distribution. This approach determines the actual daily radiation dose received by the patient's organs at risk (OARs) and evaluates its impact on radiation-induced complications. Some issues are needed to improve before publishing as follows:
- In manuscript, authors only analyzed 821 CBCT scans from 20 patients, with a relatively small sample size. The smaller sample size may limit the generalizability and statistical validity of the research results. Please clarify the limitations of the sample size in the discussion section.
- The dose volume relationship of the bladder is exponential, while the dose volume relationship of the rectum is likely linear. Please add discussion on the possible reasons for these differences.
- Acertain correlation between the dosage parameters of the bladder and rectum and clinical toxicity, but some results did not reach statistical significance. Is it limited by patients number? Integrating Normal Tissue Complication Probability (NTCP) models could facilitate a quantitative comparison, providing a more robust analysis of how these variations influence the likelihood of adverse effects.
- In table 1,Dmeanis much larger than Dmean* X fractions. Why?
- Previous studies have often refrained from implementing daily Cone-Beam Computed Tomography (CBCT) scans due to concerns regarding the additional radiation exposure to patients. Balancing the benefits of accurate tumor targeting against the potential risks of increased radiation exposure is essential. Evaluation of the necessity and frequency of CBCT scans should be conducted in the discussion section.
- Unlike simulation CT images, CBCT images frequently suffer from inferior quality, requiring meticulous correction of Hounsfield Unit (HU) values to ensure the accuracy of dose calculations. While Reference 8 in this manuscript discusses such corrections, it is specific to Elekta equipment, whereas the present study employs Varian systems. Given that different models and manufacturers may utilize varying imaging parameters, which can significantly influence dose calculation accuracy, it is essential to provide a comprehensive description of the correction methods adapted to the specific equipment used in this study. Such detailed documentation would not only enhance the credibility of the findings but also improve their reproducibility for future research.
Author Response
Dear reviewer,
thank you very much for the insightful and critical comments. Your remarks helped to increase the quality of the text. Please find the detailed responses to your comments below.
Comment 1: In manuscript, authors only analyzed 821 CBCT scans from 20 patients, with a relatively small sample size. The smaller sample size may limit the generalizability and statistical validity of the research results. Please clarify the limitations of the sample size in the discussion section.
Response 1: We agree with your comment and acknowledge that the generalizability of this study is limited by its retrospective design and small patient cohort, despite the inclusion of more than 800 CBCT scans. To address this comment, we have modified our remarks regarding the limitations of this study (please see page 12, lines 352-359).
Comment 2: The dose volume relationship of the bladder is exponential, while the dose volume relationship of the rectum is likely linear. Please add discussion on the possible reasons for these differences.
Response 2: Increased rectal volumes shift larger parts of the rectum within the horizontal irradiation plane and towards the PTV. This reduces the distance to the PTV, so that larger parts of the organ reach the high-dose area around the PTV. Based on our results, this effect is subject to a linear relationship. In contrast, increased bladder volumes cause larger parts of the organ to move out of the horizontal irradiation plane beyond the upper field limit. This effect can influence the dose exposition even more than a mainly horizontal movement as seen regarding the rectum. This might explain the exponential dose volume relationship of the bladder.
To discuss this issue, we added these remarks to the discussion of the dose volume relationship of the bladder (please see page 11, lines 293-297) and the rectum (please see page 11, lines 306-309).
Comment 3: A certain correlation between the dosage parameters of the bladder and rectum and clinical toxicity, but some results did not reach statistical significance. Is it limited by patient numbers? Integrating Normal Tissue Complication Probability (NTCP) models could facilitate a quantitative comparison, providing a more robust analysis of how these variations influence the likelihood of adverse effects.
Response 3: As you rightfully mentioned above, the results of this study may be limited by the sample size and its effect on the statistical results. However, the presentation of a correlation is also complicated by the low observed toxicity rates; in our cohort there were no severe toxicities and thus no drastic deviations in the dosimetric parameters. Furthermore, the fluctuation of the daily dosimetric parameters, as shown in Figure 2, complicate the evaluation. We agree that NTCP models may predict the impact of the organ’s daily dose exposure on treatment-related toxicities. However, whether such models can be based not only on individual planning computed tomography scans but also on CBCT scans needs to be investigated in further studies.
Comment 4: In table 1,Dmean is much larger than Dmean* X fractions. Why?
Response 4: Two different methods were used to assess the deviation between actual and planned OAR dose exposure. The upper half of table 1 compares the average dosimetric parameters of the planned/initial summation plan with the CBCT-volume summation plan. Using this method, each fractions’ volume structures are incorporated into the planning computed tomography. This is why average Dmean and Dmax values are higher, because doses are compared relative to the total dose.
In the lower half of Table 1, the average dosimetric parameters of the individual RT fractions are compared with the average dosimetric parameters of the planned/initial summation plan, which was adjusted to the daily fraction dose. The comparison is therefore relative to the daily dose.
As discussed, these two methods each have different advantages and disadvantages, resulting in different results (page 11, lines 262-279). For clarification, we have added a paragraph explaining the two different methods (please see page 3f., lines 133-139).
Comment 5: Previous studies have often refrained from implementing daily Cone-Beam Computed Tomography (CBCT) scans due to concerns regarding the additional radiation exposure to patients. Balancing the benefits of accurate tumor targeting against the potential risks of increased radiation exposure is essential. Evaluation of the necessity and frequency of CBCT scans should be conducted in the discussion section.
Response 5: We understand the concern about daily CBCT imaging. However, our clinic recognizes that although CBCT involves additional radiation exposure, optimizing positioning is more important than the additional radiation exposure, since PTV margins and thus toxicity rates can be reduced. This is supported by several studies.
To discuss the implementation of daily CBCT image guidance we have added a paragraph and the respective literature references to the discussion (please see page 12, lines 339-344).
Comment 6: Unlike simulation CT images, CBCT images frequently suffer from inferior quality, requiring meticulous correction of Hounsfield Unit (HU) values to ensure the accuracy of dose calculations. While Reference 8 in this manuscript discusses such corrections, it is specific to Elekta equipment, whereas the present study employs Varian systems. Given that different models and manufacturers may utilize varying imaging parameters, which can significantly influence dose calculation accuracy, it is essential to provide a comprehensive description of the correction methods adapted to the specific equipment used in this study. Such detailed documentation would not only enhance the credibility of the findings but also improve their reproducibility for future research.
Response 6: We agree with the reviewer’s comment that gray scale values of CBCT scans must be converted to physical density, to calculate the actual dose distribution based on each CBCT as precisely as possible. However, we disagree with the statement that the methodology presented by Richter et al. is specific to Elekta equipment. The calculation of the density curve for treatment planning is essentially identical for both manufacturers. By using real CBCT data and the respective software to calculate the calibration curve, it is defined specifically for the device in use. This is supported by the paper from Pearson et al. (Reference 6), who created a calibration curve for Varian equipment and showed that it can be used for CBCT-based dose calculations. We have added the reference to the paragraph (please see page 3, line 115).
We hope that the proposed revisions have effectively addressed your comments. Should there be any further inquiries or issues that require clarification, we would be more than happy to provide additional information.
With kind regards on behalf of the authors
Marc Petrikowski
Kai Kröger
Hans Theodor Eich